# Electronically Adjustable Grounded Memcapacitor Emulator Based on Single Active Component with Variable Switching Mechanism

**Predrag B. Petrović**

Faculty of Technical Sciences, University of Kragujevac, Svetog Save 65, 32000 Čačak, Serbia;
predrag.petrovic@ftn.kg.ac.rs

**Abstract:** New current mode grounded memcapacitor emulator circuits are reported in this paper, based on a single voltage differencing transconductance amplifier-VDTA and two grounded capacitors. The proposed circuits possess a single active component matching constraint, while the MOS-capacitance can be used instead of classical capacitance in a situation involving the simulator working within a high frequency range of up to 50 MHz, thereby offering obvious benefits in terms of realization utilising an IC-integrated circuit. The proposed emulator offers a variable switching mechanism—soft and hard—as well as the possibility of generating a negative memcapacitance characteristic, depending on the value of the frequency of the input current signal and the applied capacitance. The influence of possible non-ideality and parasitic effects was analysed, in order to reduce their side effects and bring the outcome to acceptable limits through the selection of passive elements. For the verification purposes, a PSPICE simulation environment with CMOS 0.18 μm TSMC technology parameters was selected. An experimental check was performed with off-the-shelf components-IC MAX435, showing satisfactory agreement with theoretical assumptions and conclusions.

**Keywords:** current-mode; memcapacitor emulator circuit; VDTA; grounded passive components; soft and hard switching; simulation

## 1. Introduction

In 1971 [1], Leon Chua introduced the memristor into theory as the fourth passive element to describe the missing relationship between magnetic flux and electric charge. It has found its application in various fields, such as RRAM (resistive random access memory), CNN (cellular neural networks), neuromorphic and soft computing. Chua's elementary circuit element quadrangle was extended to include higher order elements: in [2] a generalized model for the memcapacitive and meminductive systems was defined. The memcapacitor was defined as [2–4]:

$$
\begin{aligned}
v_C(t) &= C_M^{-1}(x, q, t)q(t), \\
\dot{x} &= f(x, q, t)
\end{aligned}
\tag{1}
$$

where $q(t)$ is the total charge on the memcapacitor at time $t$, $v_c(t)$ is the corresponding voltage, and $C_M^{-1}(x, q, t)$ is an inverse memcapacitance. The above formula of memcapacitor can be converted to [2]:

$$
v_C(t) = C_M^{-1}\left[\int_{t_0}^{t} q(\tau)d\tau\right]q(t) = C_M^{-1}(\sigma)q(t)
\tag{2}
$$

known as a charge-controlled memcapacitor. Here, $\sigma$ is the integral of $q$ with respect to time $t$. Like the meminductor, the memcapacitor is a memory element which retains

memory of the past dynamics [5–8], providing the connection between the flux and time integral of the charge, whereby its capacitor value depends on history, since the memcapacitor features a non-linear capacitor change. The memcapacitor gives a pinched hysteresis loop in charge and a voltage plain [9], and can be built in both voltage-controlled and charge-controlled topologies. It is precisely owing to such non-linear characteristics that the memcapacitors find application in realization of the chaotic (due to their high non-linearity, they exhibit complex dynamics) and hyperchaotic circuits (chaotic oscillators), modelling of neural networks, neuromorphic circuits and memory devices—considering that such circuits can store information without the need for a power source [10].

In the period since it was first defined as a new element, the memcapacitor provoked huge attention among researchers, primarily because a memcapacitor (and a meminductor) is not easily available, so that over the past decade there were many solutions which appeared based on the emulation of the theoretically described performances. Such solutions are primarily based on either the use of active blocks with corresponding passive circuit elements [9,11–21], or on the use of memristors and mutators [4,22–25]—they could transform a non-linear device's (memristor) characteristics to memcapacitor characteristics. Many of the described solutions require the use of a large number of elements for their implementation, and accordingly require a large space (large silicon area) within the integrated implementation—which makes them unsuitable for the realization of analogue integrated circuits. Such solutions most often require high consumption, while introducing additional non-idealities and limiting the bandwidth in which the projected emulation circuits can be used. All of the above was the main inspiration and the reason why this paper offers a completely new and original approach in the implementation of memcapacitor emulation circuits, based on the application of only one active block—the voltage differencing transconductance amplifier (VDTA) that has proven its good performance in many applications related to analog signal processing.

In [26] the incremental memcapacitor is emulated using a single multiplier, four opamps—OA, resistors, capacitors, and a current-controlled current source—while authors of [15] propose a very simple operational transconductance amplifier (OTA)-based memcapacitor emulator that does not comprise a mutator circuit. Although this circuit is simple in concept, it does not provide high performance, either in terms of cut-off operating frequency or in terms of consumption and controllability. A new general model for the design of a charge-controlled floating memcapacitor was presented in [12], using an active block DXCCDITA (dual X current conveyor differential input transconductance amplifier), two grounded capacitors and one resistor. The proposed emulator circuits offer the control over memcapacitance through switching operation between incremental and decremental mode of operation. The papers [3,20,21] describe a compact charge-controlled emulator consisting of off-the-shelf devices, based on the usage of CCII (second generation current conveyors) and analog multipliers (AM). The same base for realization is used by the authors of [9] (using a smaller number of CCIIs), while the paper [22] gives a good overview of some well-known implementations of memcapacitor emulation circuits that are realized on the basis of active blocks and mutator circuits. Two simple memcapacitor emulators were developed using CCIIs as active circuit elements in [11] with capability to electronically control values of obtained memcapacitivity and exhibiting a strong pinched hysteresis loop between the charge and voltage. Establishing the relation between memcapacitance and the Miller effect in voltage amplifiers (which is present in amplifiers with a parasitic or physically implemented capacitance connecting the input and output terminals), the emulator circuits in [4] use a two-state memristor and some passive elements for obtaining the desired characteristics. The proposed solution was implemented in a field-programmable analog array. Using a floating current source (FCS) structure the authors of [14] propose emulation circuits which enable the current mode of operation and low power consumption, but demand additional input voltage signal in order to obtain the memcapacity characteristics. The solution described in [13,19] is based on multi-output operational transconductance amplifiers (MO-OTA), two capacitors, two resistors and an

analog multiplier. The charge value of the memcapacitor can be tuned by varying the transconductances of the OTAs. A circuit for emulating a memcapacitor and meminductor of very simple structure, based on the use of only one active block-voltage differencing current conveyor VDCC, was proposed in [16]. Similarly, in [23,24] a new mutator making use of a varactor diode with variable capacitance is proposed to achieve the transformation among different mem-elements.

A new, very simple charge-controlled grounded memcapacitor emulation circuit, based on a voltage differencing transconductance amplifier-VDTA and only two grounded capacitors, which makes the proposed configuration very suitable for the production in the form of monolithic ICs, is proposed in this paper. Using this structure, the proposed emulation circuit does not utilize a multiplier or a mutator circuit and offers the possibility of adjusting the charge value of the emulator circuit by changing the transconductance gain through the biasing voltage of the VDTA. In addition to the above advantages, the circuit offers the possibility of choosing the switching mechanism—soft or hard, by choosing the value of the capacitance used, or the frequency of the current excitation signal. The change of the operating frequency and the amplitude of the input current signal affects the width of the pinched hysteresis loop, while by selecting the input voltage connection, the negative memcapacitance characteristic can also be realized [27]. The emulator circuits with hard switching behavior that can be used in the application of spiking and bursting neuron circuit, in practice exhibit two states: high and low memcapacitance. All solutions known so far are limited in terms of operating frequency when compared to the emulator circuit proposed here, enabling scalability, efficiency, better bandwidth performance and low energy consumption. In addition to the above, the VDTA used does not generate additional imperfections in terms of voltage and current gain, unlike emulator circuits based on the use of current-conveyers [3,9,11,20,21], which are hampered by limitations in terms of slew rate, bandwidth and dynamic ranges. The MOS-capacitor is also employed rather than the standard capacitor to operate suitably in the high-frequency range.

For the purpose of validation of the proposed memcapacitor emulator, the simulation using PSPICE environment for the CMOS implementation (depicted in Figure 1b) was performed within a wide frequency range of up to 50 MHz, while experimental verification was performed using commercially available components. The results obtained completely confirmed the theoretical assumptions. According to the currently available literature, the proposed emulation circuit is the first of its kind based on the use of only one VDTA and demands the smallest number of transistors for practical realization.

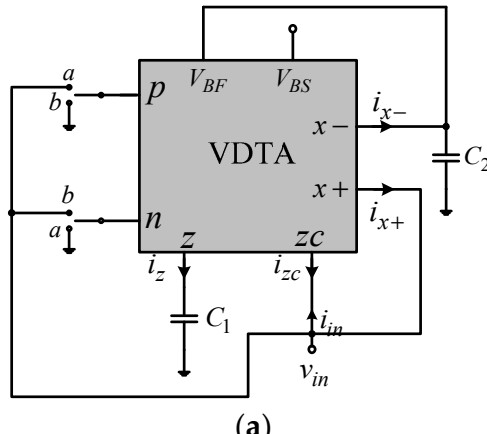

(a)

**Figure 1.** *Cont.*

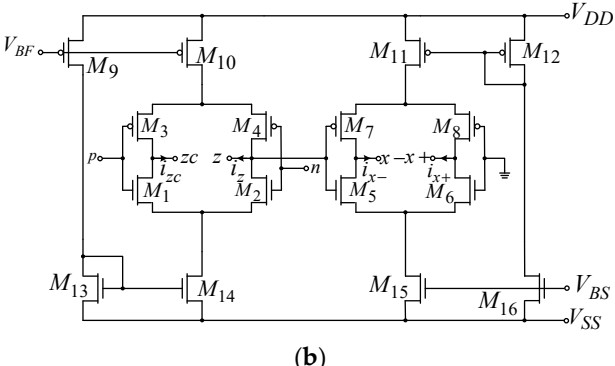

**(b)**

**Figure 1.** (**a**) Proposed memcapacitance emulator, (**b**) CMOS implementation of voltage differencing transconductance amplifier (VDTA).

## 2. Proposed Emulator Circuits

Figure 1a shows the proposed memcapacitance emulator. The functional relations (current-voltage) between the various ports of VDTA are given by Equation (3) [28,29]:

$$
\begin{bmatrix} i_p \\ i_n \\ i_z \\ i_{zc} \\ i_{x+} \\ i_{x-} \end{bmatrix} = \begin{bmatrix} 0 & 0 & 0 \\ 0 & 0 & 0 \\ g_{mF} & -g_{mF} & 0 \\ -g_{mF} & g_{mF} & 0 \\ 0 & 0 & g_{mS} \\ 0 & 0 & -g_{mS} \end{bmatrix} \begin{bmatrix} v_p \\ v_n \\ v_z \end{bmatrix}
\tag{3}
$$

where, $g_{mF}$ and $g_{mS}$ are trans-conductance gains of VDTA (two dependent transconductance gain stages) [29]. Its gains can be controlled electronically by biasing voltage $V_{BF}$ and $V_{BS}$. A copy of the current $i_z$ is available at the $zc$-terminal ($i_{zc}$); $z$ and $zc$, i.e., positive and negative output of first OTA section are auxiliary high impedance ports [29]—the suggested solutions used the $zc$ (zero copy) port, which fully utilized the potentials of VDTA as an active circuit [28,30].

Figure 1b shows the possible CMOS realization of a slightly modified VDTA (it is a six-terminal block). The modification in relation to the VDTA circuit used in [10,29,31–34] is reflected in the use of the $M_9$ (the gate of the transistor $M_9$ is used as the connecting point to introduce control-biasing voltage $V_{BF}$ in the proposed circuits) PMOS transistor gate as a control node, which is conditioned by the polarity of the generated voltage on the capacitor $C_2$ (Figure 1a)—not as usual, via the $n$-channel transistor (transistor $M_{13}$), which is a standard implementation in all VDTA applications. This change avoids the need for a special multiplication circuit, all with the aim of realizing the memcapacitive function of the emulation circuit.

As we can see in Figure 1b, two Arbel–Goldminz (AG) transconductance cells [35] form the VDTA, so that the $g_{mF}$ and $g_{mS}$ value is determined by the output transistor transconductance, which can respectively be approximated as [29]:

$$
g_{mF} \cong \left(\frac{g_1 g_2}{g_1+g_2}\right) + \left(\frac{g_3 g_4}{g_3+g_4}\right) = \sqrt{\mu_n C_{oxn}(W/2L)_9}\left(V_{DD} - V_{BF} - |V_{Tp}|\right)\left(\frac{\sqrt{\mu_n C_{oxn}(W/L)_1 (W/L)_2}}{\sqrt{(W/L)_1}+\sqrt{(W/L)_2}} + \frac{\sqrt{\mu_p C_{oxp}(W/L)_3 (W/L)_4}}{\sqrt{(W/L)_3}+\sqrt{(W/L)_4}}\right)
$$

$$
g_{mF} \cong K_F\left(V_{DD} - V_{BF} - |V_{Tp}|\right)
$$

$$
g_{mS} \cong \left(\frac{g_5 g_6}{g_5+g_6}\right) + \left(\frac{g_7 g_8}{g_7+g_8}\right) = \sqrt{\mu_n C_{oxn}(W/2L)_{16}}\left(V_{BS} - V_{SS} - V_{Tn}\right)\left(\frac{\sqrt{\mu_n C_{oxn}(W/L)_5 (W/L)_6}}{\sqrt{(W/L)_5}+\sqrt{(W/L)_6}} + \frac{\sqrt{\mu_p C_{oxp}(W/L)_7 (W/L)_8}}{\sqrt{(W/L)_7}+\sqrt{(W/L)_8}}\right)
$$

$$
g_{mS} \cong K_S\left(V_{BS} - V_{SS} - V_{Tn}\right)
$$

$$\tag{4}$$

where $\mu$ is the effective carrier mobility, $C_{ox}$ is the gate-oxide capacitance per unit area, and $W$ and $L$ are the effective channel width and length of the $i$-th MOS transistor, respectively. On the basis of such dependency, Equation (2), it follows that the temperature dependence of VDTA is related to temperature dependence of $g_m$, and depends upon the absolute magnitude of $V_T$ (threshold voltage), whereas the approximate absolute change in $V_T$ is $-2.4 \, \text{mV}/^\circ\text{C}$ [29]. The temperature dependence of mobility is $\mu(T) = \mu(T_0)(T/T_0)^{-1/5}$. While the threshold voltage decreases with the temperature in the order of 0.24%, the mobility decreases with the temperature in the order of 1.5%. Obviously, the mobility decrease is dominant on the $g_m$ and $g_m$ decreases with the increasing temperature [29].

The voltage across the capacitance $C_2$ is used as the control voltage $V_{BF}$, which is shown in Figure 1a. On the basis of the relation defined with (1), for the voltage at the external capacitor $C_1$, we obtain:

$$v_{C_1}(t) = v_z(t) = \frac{1}{C_1} \int i_{in}(t) dt = \frac{1}{C_1} q(t) \tag{5}$$

where $q(t)$ is charge of the input current signal. Consequently, the voltage on the capacitor $C_2$ is defined as:

$$v_{C_2}(t) = -\frac{g_{mS}}{C_1 C_2} \int q(t) dt = -\frac{g_{mS}}{C_1 C_2} \sigma(t) = V_{BF} \tag{6}$$

where $\sigma(t)$ is time integral of charge. The voltage $V_{BF}$, based on Equation (4), determines the value of transconductance $g_{mF}$:

$$g_{mF}(t) = K_F \left[ V_{DD} - \left( -\frac{g_{mS}}{C_1 C_2} \sigma(t) \right) - |V_{Tp}| \right] \tag{7}$$

Since the output port $x+$ of the VDTA is connected to the $zc$ port, we can write:

$$i_{in}(t) = i_{x+}(t) = g_{mS} v_z(t) = \frac{g_{mS}}{C_1} q(t) \tag{8}$$

Also, the input port $p$ or $n$ of the VDTA is connected to the $zc$ port (switches position $a$ or $b$). As indicated in Figure 1a, if port $p$ connects with port $zc$ (switching in position $a$), port n is then grounded, and vice versa, if port $n$ is connected to port $zc$, port $p$ is grounded (switching in position $b$). In this way, it is ensured that one of the differential inputs of the first AG cell of the VDTA is always grounded. For this reason we can define the input current as:

$$i_{in}(t) = \pm g_{mF} v_{in}(t) \tag{9}$$

The sign in Equation (9) depends on the switch's position (position $a$ or $b$)—in accordance with functional relations defined in Equation (3), defining the mode of operation of the proposed memcapacitance emulator circuits. It follows that:

$$g_{mS} v_z(t) = \pm g_{mF} v_{in}(t) = \frac{g_{mS}}{C_1} q(t)$$
$$v_{in}(t) = \pm \frac{g_{mS}}{g_{mF} C_1} q(t) = \pm \frac{g_{mS}}{K_F \left[ V_{DD} + \frac{g_{mS}}{C_1 C_2} \sigma(t) - |V_{Tp}| \right] C_1} q(t) \tag{10}$$

Finally, the equivalent inverse memcapacitance is given as:

$$C_M^{-1}(t) = \pm \frac{g_{mS}}{K_F \left[ V_{DD} + \frac{g_{mS}}{C_1 C_2} \sigma(t) - |V_{Tp}| \right] C_1} \tag{11}$$

The obtained Equation (11) indicates that the presented memcapacitance emulator introduces the control transconductance parameter $g_{mS}$ of VDTA, in addition to frequency and amplitude value of the input current signal. The realized functional dependence provides an additional control parameter which can regulate the apex of the pinched

hysteresis loop, while also enabling a smooth loop at high frequencies without the need to change the $f$ and amplitude of the input signal.

By selecting the port of VDTA to which the input voltage of the proposed circuit for emulation will be connected, the shape of the characteristic of the realized memcapacitor is determined, i.e., whether the pinched hysteresis loop will pass through 1 and 3, respectively 2 and 4 quadrants of the $v$-$q$ characteristic (direct or inverting memcapacitance)—the negative memcapacitance.

Whether the circuit in Figure 1a will have a soft or hard switching characteristic depends on the position of the breakpoints on the pinched hysteresis characteristic, which are determined by the moment when the charge on the proposed emulator circuits reaches the maximum value. These points are determined by the moment when the input current signal reaches the transition point—the so-called zero-crossing (assuming the input current is defined as $I_m\cos\omega t$), and provided that the value of the voltage is as small as possible, i.e., converges to zero [36,37]. In this way, the transient characteristic of the realized memcapacitance becomes closer to the axis $q$, and the circuit realizes the hard switching characteristic. The value of the input voltage at the moment when there is a change in the direction of movement of the operating point on the transient characteristic, based on the above assumptions is determined as:

$$
\begin{aligned}
v(t_1) &= \pm\frac{g_{mS}I_m}{K_F\left[V_{DD}-\left|V_{Tp}\right|\right]\omega C_1} \to 0 \\
\frac{\omega K_F C_1}{g_{mS}I_m}&\left[V_{DD}-\left|V_{Tp}\right|\right] \gg 1
\end{aligned}
\tag{12}
$$

By choosing the value of the frequency of the current excitation signal or the size of the capacitor (Equation (12)), it is possible to change the behavior of the proposed emulator circuit-switching characteristics.

*Non-Ideal and Parasitic Analysis*

In real practical conditions, the VDTA possesses non-ideal gains-tracking errors, for which reason the Equation (11) can be reformulated as:

$$
C_M^{-1}(t) = \pm\frac{\beta_S g_{mS}}{\beta_F K_F\left[V_{DD}+\frac{\beta_S g_{mS}}{C_1 C_2}\sigma(t)-\left|V_{Tp}\right|\right]C_1}
\tag{13}
$$

where $\beta_F$ and $\beta_S$ are, respectively, the defined tracking errors of the first and second stages of VDTA-Figure 1b. Typical values of these parameters (we can observe them as non-ideal transconductance gains of VDTA stages) are in the range of 0.9 to 1, with an ideal value of 1. Generally, these tracking factors remain constant and frequency-independent within low to medium frequency ranges [29]. Based on Equation (13), it can be concluded that the proposed emulator shows the magnitudes of all sensitivity values—normalized passive and active sensitivities are equal to or less than unity in magnitude. It follows that the proposed circuit offers low passive and active sensitivities.

The transconductances of OTA cells in VDTA are a frequency-dependent parameter-determining factor and its bandwidth limitation can be described by a single pole model [10]. Considering this, the equivalent memcapacitance of the proposed emulator circuits in Figure 1a can be defined (re-written), respectively, as follows:

$$
C_M^{-1}(t) = \pm\frac{\beta_S g_{mS0}\omega_S(s+\omega_F)}{\beta_F K_{F0}\omega_F(s+\omega_S)\left[V_{DD}+\frac{\beta_S\omega_S g_{mS0}}{(s+\omega_S)C_1 C_2}\sigma(t)-\left|V_{Tp}\right|\right]C_1}
\tag{14}
$$

where $g_{mS0}$ is the transconductance gain and $K_{F0}$ is the gain factor of Arbel–Goldminz cells at zero (low) frequency, while $\omega_F = 1/\tau_F$ and $\omega_S = 1/\tau_S$ are the corresponding pole frequencies. Here, $\tau_F$, and $\tau_S$ are delays corresponding to pole frequencies, respectively. In general, the values of this pole will depend on practical implementation of the VDTA [29].

The bandwidth of VDTA can be improved by inserting a compensation resistor $R$ at port $p$, and one voltage buffer realized with a pair of MOS transistors in CMOS structure shown in [38]; in this way. we can change the transconductance gain by resistance $R$, and in this way, the bandwidth of AG cell can be changed as well, since it depends on $g_m$ ($g_m = g_{m0}/(1 + g_{m0}R)$) [10].

The parasitic resistances $R_p$, $R_n$, $R_z$, $R_{zc}$, $R_{x+}$ and $R_{x-}$, and the parasitic capacitances $C_p$, $C_n$, $C_z$, $C_{zc}$, $C_{x+}$ and $C_{x-}$ [39] appear in parallel at the corresponding terminals $p$, $n$, $z$, $zc$-, $x+$ and $x$-. In the circuit (model) of the ideal VDTA, all the listed parasitic resistances can be considered to be infinitely large, while all parasitic capacitances are approximately equal to zero. These parasitics are modelled as a shunt $R$-$C$ network at all the terminals with external circuit elements [31]. In the situation when the above described parasitic components is taken into consideration, we are in a position to analyze the operation of the proposed emulator circuits in a high-frequency environment (in this situation we can neglect parasitic port resistances). We can conclude that:

$$C_M^{-1}(t) = \pm \frac{\beta_S g_{mS}}{\beta_F K_F \left[ V_{DD} + \frac{\beta_S g_{mS}}{(C_1+C_z)(C_2+C_{x-})}\sigma(t) - \left|V_{Tp}\right| \right](C_1 + C_z)} \tag{15}$$

On the base of the obtained relation, in order to preserve the performance of the proposed emulation circuits, the external capacitors $C_1$ and $C_2$ must be chosen in the way to satisfy $C_1 \gg C_{zc}$ and $C_2 \gg C_{x-}$. Owing to this setting, the parasitic capacitance effects can be absorbed at working frequencies. In order to reduce the impact of the parasitic resistances, the values of $C_1$ and $C_2$ must be chosen as $1/sC \ll R_{zc}$ and $1/sC_2 \ll R_{x-}$ [28].

## 3. Simulation and Experimental Results

The emulator circuits proposed in Figure 1a were verified using a simulation procedure in SPICE environment, with level-49 TSMC 0.18 μm CMOS process parameters. Values for $W$ and $L$, the channel width and length of transistors in the circuit in Figure 1b are defined in the same way as in [31]. An input sinusoid current signal of varying frequency and amplitude $I_m$ = 200 μA with 90° phase shift was used in the course of simulation, while the power supply was ± 0.9 V. The power consumption of the proposed emulator circuits was 0.82 mW—much lower than emulator proposed in [20]: 14.741 mW.

Figure 2a shows the simulation results on the proposed emulation circuit at an excitation current signal frequency of 10 MHz and an amplitude of 200 μA, for different values of ground capacitances $C_1$ and $C_2$. In this way, the transition from one switching mechanism to another is demonstrated. By reducing the capacitance, the bending point on the transient characteristic becomes closer to the $q$ axis, i.e., the value of the voltage corresponding to that point converges to 0, as predicted through theoretical analysis. Practically, by increasing and decreasing capacitors values, the pinched hysteresis curve area decreases and increases respectively. This happens because the linear time-invariant part of the memcapacitor is dominated by the linear time-variant part, Equation (11). The same can be achieved by changing the frequency of the input current signal.

In Figure 2b in inverting working mode, the performance check was performed for different frequencies and the amplitudes of the excitation signal, at a constant value of the capacitance used ($C_1 = C_2 = 5$ pF). With the increase of the working frequency, it is concluded that the area covered by the lobes on the $q$-$v$ plane is being reduced, which is a well-known characteristic of memcapacitive elements. With an increase in frequency, the pinched hysteresis curve area decreases, and the memcapacitor behaves like an ordinary capacitor, i.e., a linear charge-voltage curve is observed at higher frequencies—the curve moved towards linearity just like the traditional capacitor, and the variable part diminishes as the frequency of operation increases (Equation (11)). The maximum operating frequency of the proposed emulator circuits is 50 MHz, Figure 2c, as a consequence of realistically achievable capacitance values in the integrated technique, directly resulting from the size



of the capacitors used ($C_1$ and $C_2$), since they become the order of magnitude of parasitic capacitances that exist on VDTA ports [29].

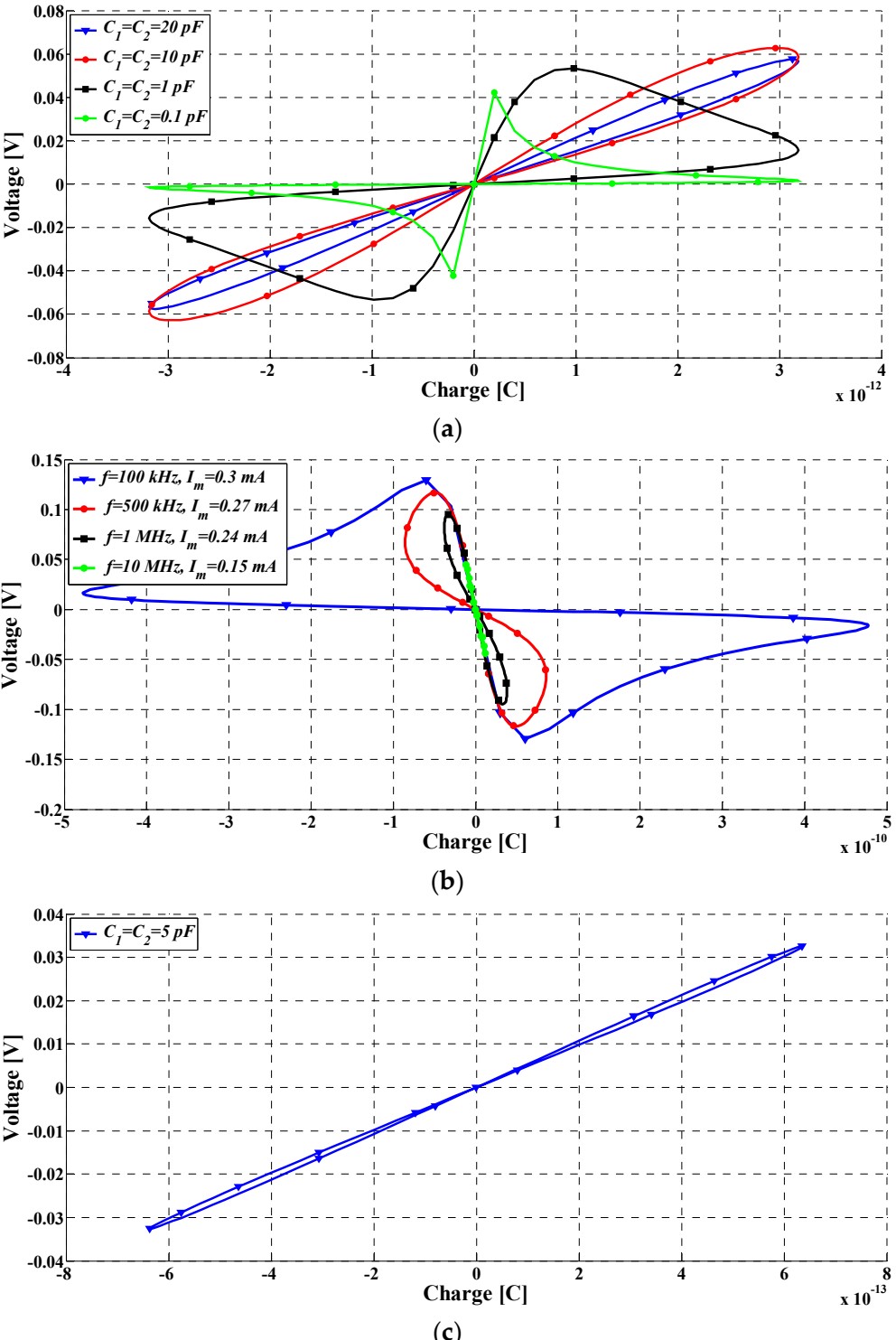

**Figure 2.** Pinched-hysteresis loops generated by the proposed charge controlled grounded memcapacitor emulator (**a**) for different capacitances, $f$ = 10 MHz; (**b**) for different frequencies and amplitude of the input current signal in inverting mode of operation, $C$ = 100 pF; (**c**) for frequency of 50 MHz and amplitude $I_m$ = 200 μA.

In order to show transient characteristic of the proposed structures in Figure 1a, the time domain response of the proposed memcapacitor emulator, in Figure 3a, the simu-

lation is repeated for a sinusoidal input current signal amplitude of 100 µA and frequency of 10 kHz. The generated voltage signal does not have a sinusoidal waveform, due to the fact that the value of memcapacitance changes with respect to time. If the frequency of the input current signal increases, the phase difference between the input voltage signal and the charge value of the memcapacitance decreases. The proposed emulation circuits, among other things, have an electronically adjustable characteristic, because the value of the charge of the circuit can be adjusted by changing the transconductance value through the bias voltages of the VDTA. For different bias voltages, the input charge–voltage relationships of the proposed memcapacitor emulator are shown in Figure 3b ($C_1 = C_2 = 50$ pF, $f = 1$ MHz, $I_m = 0.2$ mA). The temperature performance of the designed circuit subjected to three different temperatures is shown in Figure 3c based on the presented characteristics, with the proposed circuits retaining the projected properties over a wide temperature range ($C_1 = C_2 = 50$ pF, $f = 1$ MHz, $I_m = 0.25$ mA). It is also important to note that with falling temperature levels, the current flow in the memcapacitor is enhanced.

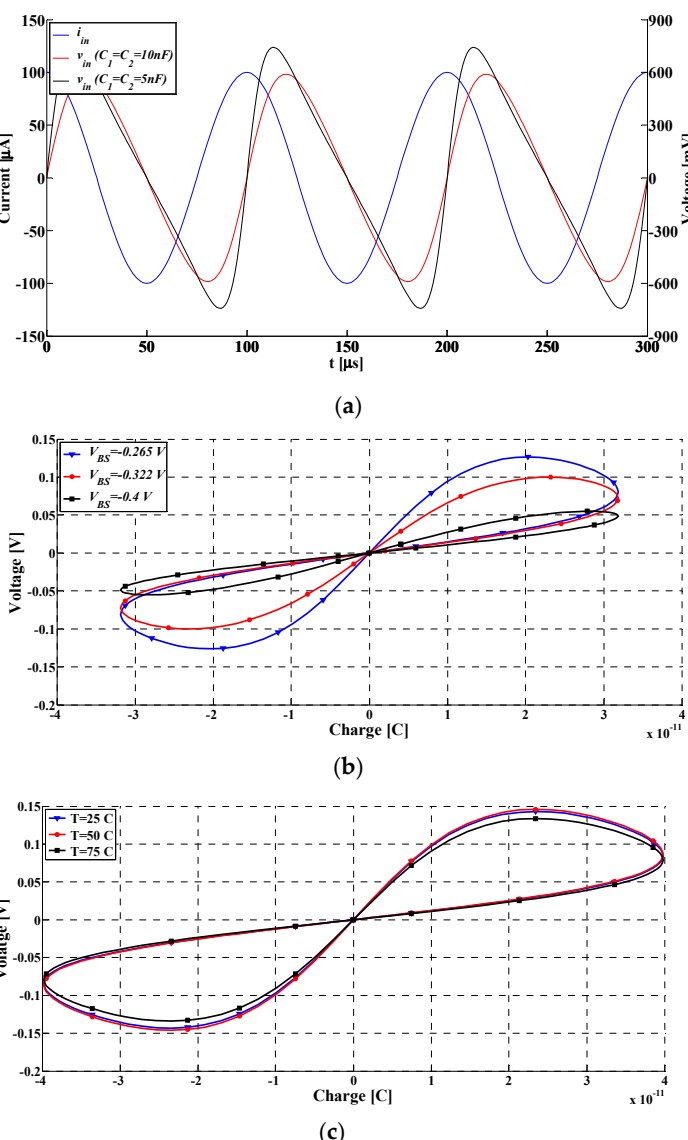

**Figure 3.** (**a**) Time-domain response of the proposed memcapacitor emulator, (**b**) the input voltage-charge relationships for the different bias voltages of VDTA ($C_1 = C_2 = 50$ pF, $f = 1$ MHz) and (**c**) hysteresis loop at various temperatures.

Figure 4 shows the variation of the emulated memcapacitance with time, having a pulse input amplitude of 0.5 mA, with a period of 1 μs and a pulse width of 0.1 μs—the non-volatile nature of the memcapacitor; the memcapacitance value remains constant even in the absence of pulse signal ($C_1 = C_2 = 1$ nF, $V_{BS} = -0.22$ V). The memcapacitance does not change when no input pulse is applied and during the pulse period, there is an abrupt change in memcapacitance value and the proposed circuit shows strong memory property between pulses. This is the confirmation of memory behavior of memcapacitor as one of its most important characteristics. In this way it is possible to illustrate the suitability of the proposed memcapacitor emulator for study and design of neuromorphic circuits provided with synaptic plasticity e.g., in particular, an LTP (long-term potentiation). This principle, inside the neuroscience field, confirms a persistent strengthening of synapses based on recent patterns of activity, and it is assumed to be one of the major mechanisms that underlies learning and memory [40]. By switching the proposed emulation circuit into the inverting mode, it is possible to provide an incremental mode of operation, since then the memcapacitance value increases from the initially negative value, Equation (11).

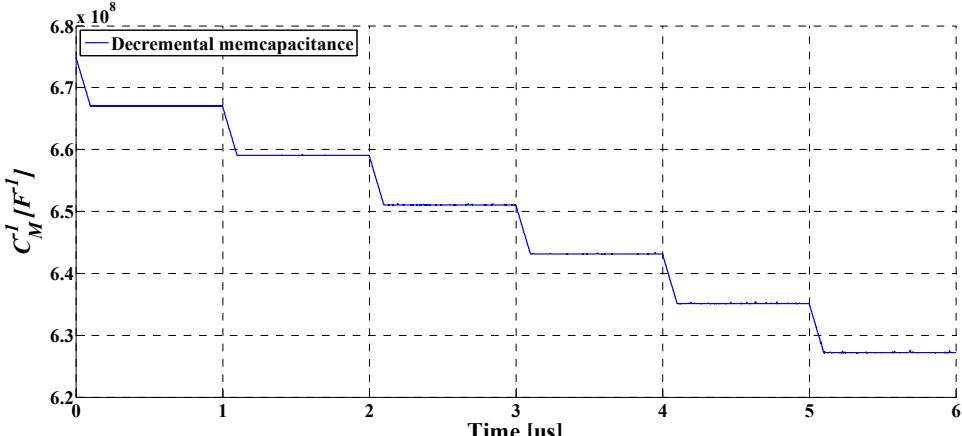

**Figure 4.** Variation of memcapacitance with time for pulse voltage having $T_{on} = 0.1$ μs for circuits in Figure 1a at 1 MHz.

### 3.1. Process Variation

Process corners like Nominal NMOS Nominal PMOS (NN), Fast NMOS Fast PMOS (FF), Slow NMOS Slow PMOS (SS), Fast NMOS Slow PMOS (FS), and Slow NMOS Fast PMOS (SF) variations are analyzed in Figure 5; hence the investigation of process variations is absolutely crucial for monolithic integration [10]. The process corners analysis is carried out at 10 MHz with $C_1 = C_2 = 10$ pF, $I_m = 0.3$ mA. Figure 5 shows that voltage flow in the case of FF mode is larger than in SS mode, as expected—a lower voltage flow in the SS process corner when compared with the FF process corner. Despite the area variation in the hysteresis loop, the proposed memcapacitor circuits exhibit a pinched hysteresis loop in all process corners. Furthermore, no offset is observed in the characteristics. In summary, the proposed circuit can be operated successfully in wide temperature ranges and different radical conditions.

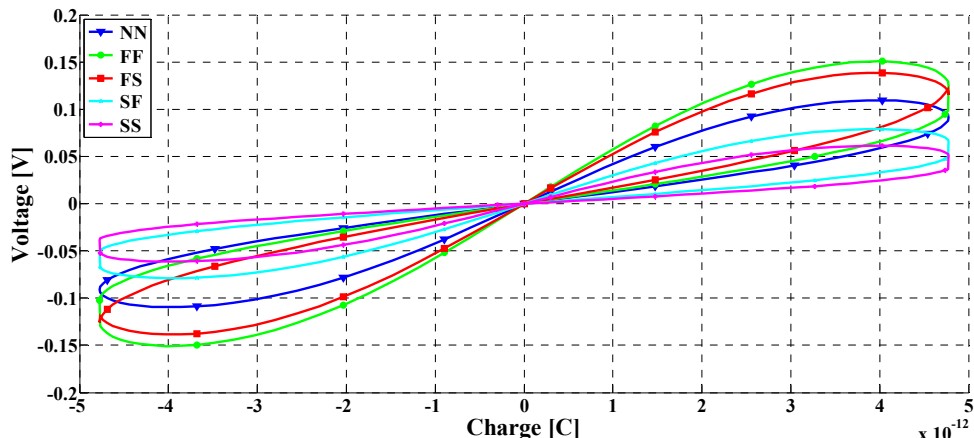

**Figure 5.** Pinched hysteresis loop variation at different process corners.

### 3.2. Experimental Results

In order to better view the performance of the proposed emulator in the light of real performances, since VDTA is not a commercially available component, the circuit shown in Figure 1a was built with off-the-shelf electronic devices. Implementation and practical measurements were performed on a discreetly implemented circuit as shown in Figure 6a, using a single MAX435 by Maxim Integrated [10,28–30,32] (the VDTA can also be realized by using single LM13700 integrated circuit [33,41], or commercially available operational transconductance amplifier (OTA/CA3080) [34]). The reason for using this component lies in the fact that it was available to the author and that he has many years of experience in the implementation of experimental setups on that platform [27,32,40].

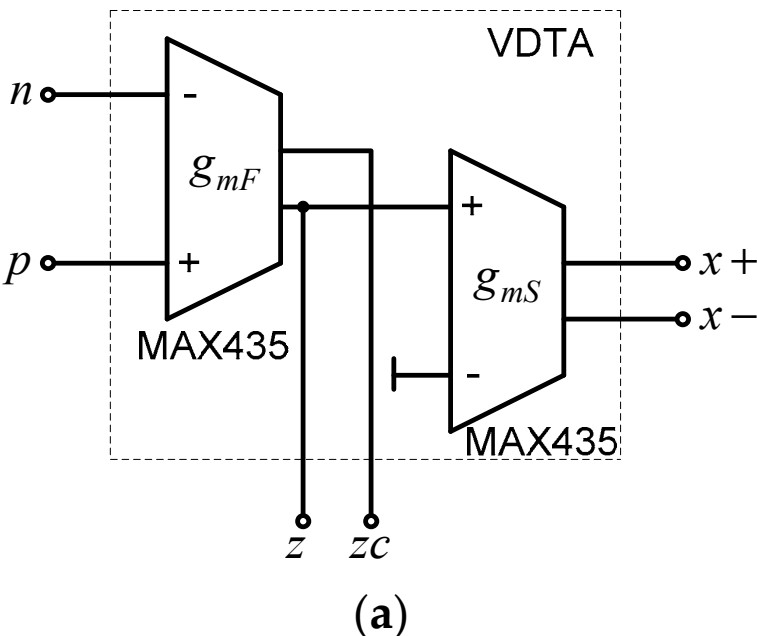

**(a)**

**Figure 6.** *Cont.*

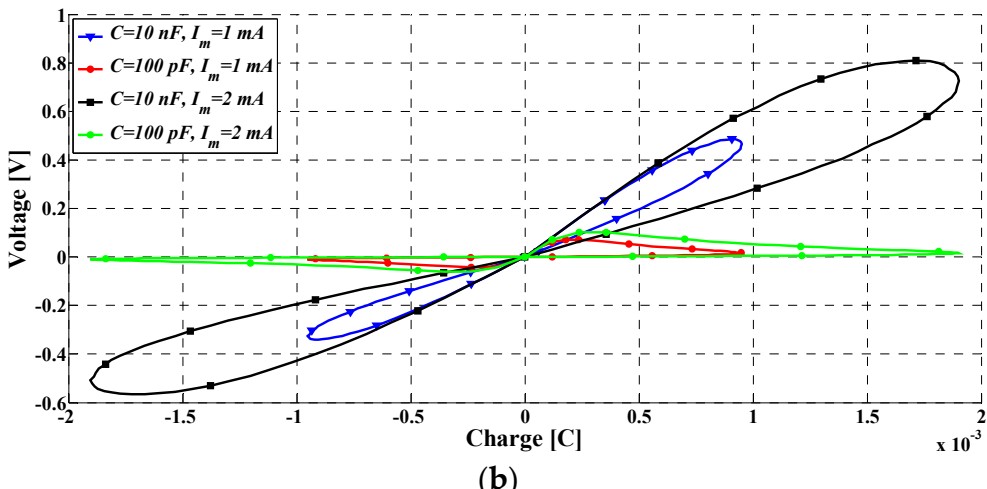

**Figure 6.** (**a**) Practical implementation of VDTA. (**b**) Experimental result of pinched hysteresis loop at 100 kHz obtained using commercial integrated circuits (ICs).

Measurements were performed using a Digilent Analog Discovery 2 board and probes, while the supply voltages were set to $\pm 5$ V, and amplitude of the input current signal varied from $I_m = 1$ mA to $I_m = 2$ mA. The pinched hysteresis loops shown in Figure 6b were obtained for the operating frequencies of 100 kHz (as a result of the limitations of available devices) and by scaling down the value of $C$. It is obvious that the obtained measurement results on experimental setup are of a similar nature to that of simulated one. It is also proved that by scaling down the capacitor value, the pinched hysteresis loop can become closer to the $q$ axis, that is, the switching mechanism changes. In the future, the proposed design will be realized in the form of an integrated circuit; however, it was currently impossible to organize it in the conditions available to the author.

The comparison is shown in Table 1, which highlights the different design and performance-related aspects of the previously reported memcapacitor emulators and of the proposed one. Comparison was performed only with emulators of memcaptivity, for which the emulation circuits which were also based on use of only one VDTA [32–34]; emulated memristive elements could not be taken into account. Such a circuit requires a smaller number of capacitors in its implementation and a different concept of processing in relation to the solution described herein. Based on the various parameters compared in Table 1, it can be concluded that the proposed emulator circuit offers the possibility of operating at the highest operating frequency in relation to all known circuits, with very low consumption, while also requiring a small number of transistors for their implementation. In addition, the proposed solution offers the possibility of working with two different switching mechanisms, soft and hard, as well as generating a negative memcapacitive characteristic, which is a significant advantage over previously known solutions. The performances for the circuits proposed in [3,4,9,11,15,21,23,24] are measured on real devices (marked with [meas]), while for all others they are results of simulations.

**Table 1.** Overview of the different design related details of the previously reported memcapacitor emulators.

| Ref. | Number of Active Comp. | Number of Floating Passive Elements | Number of Grounded Passive Elements | Power Supply | Output Range | Sim./ Exp. | Floating/ Grounded Memcapacitor Emulator | Electron. Control. | Using Memristor |
|---|---|---|---|---|---|---|---|---|---|
| [3] meas. | 5 CCII, 1 AM/3CCII, 1 AM | 1/- | 6/6 | ±10 V | 5 kHz | Both | F/G | No | No |
| [4] meas. | 2 OA | 2 | - | ±5 V | 50 Hz | Both | G | No | Yes |
| [9] meas. | 2 CCII, 1 AM | 2/2 | 2/4 | ±10 V | 25 kHz | Both | F/G | No | No |
| [11] meas. | 2 CCII, 1 AM | 1 | 3 | ±10 V | 2 kHz | Both | G | Yes | No |
| [12] | 1 DXCCDITA | - | 3 | ±1.25 V | 1 MHz | Sim. | F | No | No |
| [13] | 2 MO-OTA, 1 AM | 1 | 3 | ±0.9 V | 10 Hz | Sim. | G | Yes | No |
| [14] | 1 FCS, 1 AM | 1 | 1 | ±0.9 V | 10 Hz | Sim. | G | No | No |
| [15] meas. | 1 OTA, 1 AM | 1 | 1 | ±10 V | 500 Hz | Sim. | G | No | No |
| [16] | 1 VDCC | - | 1 | ±0.9 V | 700 kHz | Sim. | F | No | Yes |
| [17] | 1 DVCCTA | - | 3 | ±0.9 V | 1 MHz | Sim. | G | No | No |
| [18] | 2 VDCC | 1 | 3 | ±0.9 V | 10 kHz | Both | G | Yes | Yes |
| [19] | 2 MO-OTA | 1 | 3 | ±0.9 V | 1 kHz | Sim. | F | Yes | No |
| [20] | 2 CCII, 1 OTA | - | 4 | ±1.2 V | 1 MHz | Both | G | No | No |
| [21] meas. | 4 CCII, 2 AM | 6 | 2 | ±10 V | 360 kHz | Exp. | G/F | No | No |
| [23] meas. | 4 CFOA, 1 OA, 1 VD | 5 | 2 | ±15 V | 10 kHz | Both | F | No | No |
| [24] meas. | 4 CFOA, 1 VD | 3 | 1 | ±15 V | 180 kHz | Both | F | No | No |
| [25] | 1 CBTA | - | 1 | ±0.9 V | 250 kHz | Sim. | F | Yes | Yes |
| This work | 1 VDTA | - | 2 | ±0.9 V | 50 MHz | Both | G | Yes | No |

\* AM = analogue multiplier VD = varactor diode meas. = measured on real devices.

## 4. Conclusions

This paper proposes a completely new structure of current-mode grounded charge-controlled memcapacitance emulator, based on the application of VDTA. The proposed circuit is able to provide two different switching mechanisms, soft and hard, depending on the value of the capacitance used, i.e., the frequency of the current signal at the input. The paper also lays down the criteria that define the transition from one mode of work to another. In addition, the circuit is able to simulate a negative memcapacitance characteristic, which can be used in the realization of non-linear chaos oscillators and neuromorphic computing. Irrespective of the frequency and amplitude values of the input current signal, it is possible to adjust the shape of the pinched hysteresis loop through the transconductance gain of VDTA. In order to reduce the parasitic effects non-ideal analysis has been conducted, making it possible to properly perform selection of the passive circuit elements. In order to confirm the expected performance (obtained on the basis of theoretical assumptions and conclusions) of the proposed emulation circuit, variability analysis based on a variety of parameters such as process variation, capacitor variation, temperature vari-

ation, and frequency variation has been carried out. Confirmation of the proposed concept was performed through simulation and experimental verification, whereby the simulation confirmed the possibility of working at frequencies of up to 50 MHz. The proposed solution overcomes some of the limitations that existed in the implementation of previously known emulation circuits in terms of consumption, maximum operating frequency and number of used transistors in the integrated implementation.

**Funding:** This research was funded by the Ministry of Education, Science and Technological Development, Republic of Serbia, Grant No. 42009 and 172057.

**Conflicts of Interest:** The authors declare no conflict of interest.

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
