# Peer review of "Electronically Adjustable Grounded Memcapacitor Emulator Based on Single Active Component with Variable Switching Mechanism"

_electronics, doi:10.3390/electronics11010161_

Round 1

Reviewer 1 Report

The memristor emulator presented in [29], must be included in the comparison Table.

Other important relevant references are missed. For example:

Pal, I., Kumar, V., Aishwarya, N. et al. A VDTA-based robust electronically tunable memristor emulator circuit. Analog Integr Circ Sig Process 104, 47–59 (2020). https://doi.org/10.1007/s10470-019-01575-y

J. Vista and A. Ranjan, "Flux Controlled Floating Memristor Employing VDTA: Incremental or Decremental Operation," in IEEE Transactions on Computer-Aided Design of Integrated Circuits and Systems, vol. 40, no. 2, pp. 364-372, Feb. 2021, doi: 10.1109/TCAD.2020.2999919.

These must be also included in the comparison Table in order to be evident   possible contribution of the presented scheme.

Author Response

Reviewer # 1

Thank you for your generally positive attitude about my paper, and for the comments which helped me to improve the presentation and quality of my paper.

Comments: The memristor emulator presented in [29], must be included in the comparison Table.

Other important relevant references are missed. For example:

Pal, I., Kumar, V., Aishwarya, N. et al. A VDTA-based robust electronically tunable memristor emulator circuit. Analog Integr Circ Sig Process 104, 47–59 (2020). https://doi.org/10.1007/s10470-019-01575-y

  1. Vista and A. Ranjan, "Flux Controlled Floating Memristor Employing VDTA: Incremental or Decremental Operation," in IEEE Transactions on Computer-Aided Design of Integrated Circuits and Systems, vol. 40, no. 2, pp. 364-372, Feb. 2021, doi: 10.1109/TCAD.2020.2999919.

These must be also included in the comparison Table in order to be evident possible contribution of the presented scheme.

Author response: The reviewer rightly pointed out two other important references, which point to the realization of memristive elements based on the use of VDTA as an active block, which was used for the realization of the circuit for emulation of memcapacity proposed here. In accordance with the recommendation, the author included them in the reference list, while also reffering to them in the paper as guidelines that were used in the proposed implementation. However, it is not possible to use them as parts of the Table in which the comparison with other memcapacity emulators is performed (ONLY such circuits were used for comparison), since the three mentioned works emulate the memristive element, which conditions a different number of used reactive elements-capacitors, a different processing concept, and therefore the performance of these circuits. The author himself is the author of one of such emulators based on the application of only one VDTA circuit (work from 2019 in IET Circuits, Devices and Systems). In accordance with this concept of the paper, the author uses the opportunity to justify in this way why he could not fully comply with the recommendation made by the reviewer. This is another confirmation of the attractiveness of the proposed emulation circuit, as it is based on an active block that has shown its good performance in several realizations that are described in the professional literature. The proposed memcapacity emulation circuit is the first of its kind based on the use of only one VDTA. Certainly, I take this opportunity to thank the reviewer for his certainly useful suggestions and suggestions.

Reviewer 2 Report

The paper presents a memcapacitor emulator, including both simulation and some experimental results. The paper is quite well-written (though quite a bit of English editing is necessary) but I think it lacks in a few important aspects that should be improved before publication:

  1.  The concept of mem-components might not be well known to people who aren't directly involved in research about them. A few more lines of explanation on what are their basic constituent relationships that the proposed circuit is supposed to emulate would be beneficial to the average reader. In the present paper there is only line 35 to describe the desired behavior...
  2. In general, there is some lack of description about the background items used. For instance VDTA acronym is not even defined in the paper (only in the abstract), it's operation is defined in Equation (1) but then Fig. 1(b) reports a "slightly modified VDTA" but what these modifications are are not at all clear from the figure. I assume they are on the biasing, as the text seems to imply, so are they on (external) usage or (internal) structure?
  3. In Fig. 1(a) the purpose of the switches remains unclear until the end of the paper, where it is revealed that they change the sign of the memcapacitance variation. I think their intended usage should be mentioned straight away in Sect. 2. Moreover, the inputs of the first transconductance differential amplifier are always shorted together, I think some explanations on the idea that lead to such an atypical configuration might be useful. I reckon that the first stage is not really differential, just two matched but independent transconductance stages needed for the copy output of the current.
  4. I'm a bit puzzled about some of the statements in Section 3.1: corner analysis is not supposed to reveal offset in differential amplifiers, as it only applies variations between N and P, and not among N transistors themselves (or P themselves) as it would be necessary to highlight the onset of offsets. I think the discussion about offset should be improved, or removed. Nowadays there are plenty of techniques for offset reduction, those few lines add nothing, nor cite anything useful, and the effects of uncompensated offsets are not studied.
  5. About experimental results, what are the motivations that made you chose that particular component? I think the specific requirements for your particular application should be stated. Is any OTA suitable? Because the choice of a discontinued, nearly 30-year-old component like MAX435 seems quite odd. Aren't there modern alternatives that could be used?
  6. Finally, the graphs report the unit "Q" for charge. Shouldn't it be "C" (coulomb)?

Author Response

Reviewer # 2

I would like to use this opportunity to thank you for the time dedicated to my paper. The author would like to express sincere gratitude to the reviewer for contributing the remarks that have helped to improve the quality of the paper and to make it much clearer and more applicable. The new version of the paper has a much clearer layout and is easier to follow. The imprecisions that were present in the first version of the paper have now been removed, following the recommendations from the reviewer.

Comment # 1: The concept of mem-components might not be well known to people who aren't directly involved in research about them. A few more lines of explanation on what are their basic constituent relationships that the proposed circuit is supposed to emulate would be beneficial to the average reader. In the present paper there is only line 35 to describe the desired behavior...

Author response: Through the redacted, new version of the paper, the authors paid special attention to these issues, for which they thank the reviewer most sincerely. In the introductory part of the paper, additional explanations are introduced which define more precisely the emulated quantities and their functional dependence. This is explained in sufficient detail in the new version of the paper.

Comment # 2: In general, there is some lack of description about the background items used. For instance VDTA acronym is not even defined in the paper (only in the abstract), it's operation is defined in Equation (1) but then Fig. 1(b) reports a "slightly modified VDTA" but what these modifications are not at all clear from the figure. I assume they are on the biasing, as the text seems to imply, so are they on (external) usage or (internal) structure?

Author response: All the statements made on this issue by the reviewer are completely true. However, we have ensured that this is supported in the new version of the paper by additional comments and explanations that have been entered. The use of acronyms for VDTA is now accompanied by its full name in several places, not only in the abstract, so that there is no doubt on that basis. In addition, the terminology used in the paper - slightly modified VDTA, is explained in a much more precise and clear way. Namely, it refers to the fact that in the proposed MOS implementation of VDTA, the control voltage (biasing voltage) for the first OTA cell (AG cell) is introduced via a p-channel MOS transistor, and not as usual via n-channel transistor-which is a standard implementation in all VDTA applications and reference papers. The reason for this modification lies in the fact that the polarity of the generated voltage is such that it requires such a change, while not disrupting the functional connection that exists between the voltage and port currents of the VDTA used.

Comment # 3: In Fig. 1(a) the purpose of the switches remains unclear until the end of the paper, where it is revealed that they change the sign of the memcapacitance variation. I think their intended usage should be mentioned straight away in Sect. 2. Moreover, the inputs of the first transconductance differential amplifier are always shorted together, I think some explanations on the idea that lead to such an atypical configuration might be useful. I reckon that the first stage is not really differential, just two matched but independent transconductance stages needed for the copy output of the current.

Author response: As in the case of the previously made statements, the facts presented here are completely in place. As suggested by the esteemed reviewer, the function of the switch is clearly and precisely described in Section 2. The inputs of the first differential stage are not short-circuited, but one of them is always grounded, which is provided by the position of the switch. In this way, the polarity of the input differential voltage changes, threby changing the functional dependence generated by the emulator circuit. In the new version, it is presented in a transparent way and explained so that there would be no additional confusion on that issue. The input stage is really differential, whereas the second stage is practically the same except that its excitation is specified via the z port, with the second differential input being always grounded.

Comment # 4: I'm a bit puzzled about some of the statements in Section 3.1: corner analysis is not supposed to reveal offset in differential amplifiers, as it only applies variations between N and P, and not among N transistors themselves (or P themselves) as it would be necessary to highlight the onset of offsets. I think the discussion about offset should be improved, or removed. Nowadays there are plenty of techniques for offset reduction, those few lines add nothing, nor cite anything useful, and the effects of uncompensated offsets are not studied.

Author response: In order to avoid confusion on this issue, and in full accordance with the proposal given, this part has been deleted. It is completely true that today there really is a plenty of techniques for offset reduction, which will be applied depending on the integration process itself.

Comment # 5: About experimental results, what are the motivations that made you chose that particular component? I think the specific requirements for your particular application should be stated. Is any OTA suitable? Because the choice of a discontinued, nearly 30-year-old component like MAX435 seems quite odd. Aren't there modern alternatives that could be used?

Author response: The reason for using this component lies in the fact that it was available to the author and that he has many years of experience in the implementation of experimental setups on that platform. The new version of the paper states that other more modern components can be used for this purpose, but since the author's intention is to confirm the possibilities and concept of the proposed solution through experiment, the realization was based on what was available and what the author is sufficiently familiar with. The first proposed and described VDTA solution from 2013 is based on this component.

Comment # 6: Finally, the graphs report the unit "Q" for charge. Shouldn't it be "C" (coulomb)?

Author response: At this point, I really sincerely thank the reviewer for the observed omission in marking the unit that measures the charge. All diagrams have been corrected in accordance with this comment.

Reviewer 3 Report

The paper proposes a new electronic circuit for the emulation of the memcapacitor. The paper introduces the subject with a good analysis of the background and state-of-the-art; gives the basic theory of the problem;  explain the proposed circuit; and reports simulations and measurements on a real circuit based on commercial devices. The results are compared to what available in literature. The paper is complete, easy to follow, and of interest for the readers of the journal.

Here my suggestions:

  • Author stresses that the proposed circuit works well up to 50 MHz. However, this reviewer found no proof for this statement in the work. Author is encouraged to add tests that show how the circuit perform at 50 MHz and which the limitations are in this condition.
  • Most of the work and performance analysis is based on simulations. A real circuit is realized, but its analysis is limited to 100kHz (I guess since the limitations of available devices). Simulations are effective, but not all of the physical phenomena are included. Is an integrated realization of the proposed circuit planned?
  • Since most of the work is based on simulations, I suggest to specify in Table 1 if the performance of the reported circuits is measured on real devices or just simulated.

Minor: the size of the character employed in the test is not uniform.

Author Response

Reviewer # 3

At this point, the author must thank the reviewer for his more than positive attitude towards the results they came to while working on this project and the idea. Your attitude gives me additional strength and confidence that I am on the right path. Thanks again for everything, as well as for the time you have dedicated to my work, as well as more useful and good-natured comments that have enabled me significantly improve my work. The new version, at least I hope, and I deeply believe it, is much more precise in its fortifications and more comprehensible to the broader reading audience. For their part, the author have done everything within his power to respond in a more precise and clearer way to all the supplied comments. For all of this, I owe a great thank you to the reviewer.

Comment # 1: Author stresses that the proposed circuit works well up to 50 MHz. However, this reviewer found no proof for this statement in the work. Author is encouraged to add tests that show how the circuit perform at 50 MHz and which the limitations are in this condition.

Author response: In the new version of the paper, a Figure-test has been added, which confirms this possibility through a simulation test. The limitation on the maximum frequency-range, in which the proposed circuit can provide emulation of memcapacitance, is a direct consequence of the size of the capacitors used, because they become the order of magnitude of parasitic capacitances that exist on VDTA ports. All this is included in the new version of the paper with sufficient comments and explanations.

Comment # 2: Most of the work and performance analysis is based on simulations. A real circuit is realized, but its analysis is limited to 100kHz (I guess since the limitations of available devices). Simulations are effective, but not all of the physical phenomena are included. Is an integrated realization of the proposed circuit planned?

Author response: Of course, the author plans to realize the proposed design in the form of an integrated circuit - however, it is currently technically and financially impossible to organize this either at the institution where he works, or at any other institution and faculty in Serbia. If there is an opportunity to establish cooperation with one of the foreign institutions, it would speed up the solution of this problem as well. What stands as a current possibility is only the estimation of the dimensions of the layout circuit without the possibility of realizing practical measurements.

Comment # 3: Since most of the work is based on simulations, I suggest to specify in Table 1 if the performance of the reported circuits is measured on real devices or just simulated.

Author response: As suggested by the esteemed reviewer, the new version of the paper specifies whether these performances are the result of simulation tests or realistically recorded characteristics, so that on this basis, any doubts have been removed.

Minor Comment: The size of the character employed in the test is not uniform.

Author response: In the new version of the paper, the size of the character employed in the test is uniform.

Round 2

Reviewer 1 Report

My concerns have been fully addressed.

Reviewer 3 Report

I have no more comments